# School-age outcomes of children after perinatal brain injury: a systematic review and meta-analysis

Philippa Rees ![ORCID],[1] Caitriona Callan,[2] Karan Chadda,[3] Meriel Vaal,[1] James Diviney,[4] Shahad Sabti,[5] Fergus Harnden ![ORCID],[6] Julian Gardiner,[1] Cheryl Battersby ![ORCID],[7] Chris Gale ![ORCID],[7] Alastair Sutcliffe[1]

For numbered affiliations see end of article.

**Correspondence to**
Dr Philippa Rees; p.rees@ucl.ac.uk

## ABSTRACT

**Background** Over 3000 children suffer a perinatal brain injury in England every year according to national surveillance. The childhood outcomes of infants with perinatal brain injury are however unknown.

**Methods** A systematic review and meta-analyses were undertaken of studies published between 2000 and September 2021 exploring school-aged neurodevelopmental outcomes of children after perinatal brain injury compared with those without perinatal brain injury. The primary outcome was neurodevelopmental impairment, which included cognitive, motor, speech and language, behavioural, hearing or visual impairment after 5 years of age.

**Results** This review included 42 studies. Preterm infants with intraventricular haemorrhage (IVH) grades 3–4 were found to have a threefold greater risk of moderate-to-severe neurodevelopmental impairment at school age OR 3.69 (95% CI 1.7 to 7.98) compared with preterm infants without IVH. Infants with perinatal stroke had an increased incidence of hemiplegia 61% (95% CI 39.2% to 82.9%) and an increased risk of cognitive impairment (difference in full scale IQ −24.2 (95% CI −30.73 to −17.67) . Perinatal stroke was also associated with poorer academic performance; and lower mean receptive −20.88 (95% CI −36.66 to −5.11) and expressive language scores −20.25 (95% CI −34.36 to −6.13) on the Clinical Evaluation of Language Fundamentals (CELF) assessment. Studies reported an increased risk of persisting neurodevelopmental impairment at school age after neonatal meningitis. Cognitive impairment and special educational needs were highlighted after moderate-to-severe hypoxic-ischaemic encephalopathy. However, there were limited comparative studies providing school-aged outcome data across neurodevelopmental domains and few provided adjusted data. Findings were further limited by the heterogeneity of studies.

**Conclusions** Longitudinal population studies exploring childhood outcomes after perinatal brain injury are urgently needed to better enable clinicians to prepare affected families, and to facilitate targeted developmental support to help affected children reach their full potential.

Perinatal brain injuries can have wide-ranging deleterious consequences for children, families and broader society.[1–4] Over 3000 infants

## WHAT IS ALREADY KNOWN ON THIS TOPIC

⇒ Thousands of children suffer a brain injury around the time of birth every year. Many of these injuries are associated with neurodevelopmental impairment at 2 years of age. However, 2-year outcomes are not necessarily representative of later childhood outcomes and function, which are a priority for parents.

## WHAT THIS STUDY ADDS

⇒ This review provides an overview of existing evidence of childhood outcomes after perinatal brain injury. It indicates that there is some evidence of ongoing impairment throughout childhood for different types of perinatal brain injury but that there are considerable gaps in knowledge.

## HOW THIS STUDY MIGHT AFFECT RESEARCH, PRACTICE OR POLICY

⇒ This review shows the need for detailed high-quality longitudinal population studies exploring childhood outcomes after perinatal brain injury.

experience perinatal brain injury in England annually[1] and the Department of Health and Social Care (DHSC) has committed to halving the rate of perinatal brain injuries by 2030 as part of the national maternity ambition.[5] To monitor progress towards this goal, a standardised definition of perinatal brain injury was developed.[6] The degree to which this definition captures and represents true perinatal brain injuries is unclear and requires us to look beyond the neonatal period.[6]

Focusing on the childhood outcomes of infants with perinatal brain injury provides a fuller understanding of the population captured by the DHSC definition. Despite their importance to families, school-age outcomes following neonatal care have been an overlooked research priority. Neonatal studies typically focus on 2-year composite outcomes, which may mask the true neurodevelopmental burden of injuries, and are

known to be poorly predictive of future functioning.[7–10] As such, our understanding of childhood developmental trajectories after brain injuries—and whether any sequelae are fixed, stable or amenable to interventions—is limited. We therefore undertook a systematic review to explore school-age neurodevelopmental outcomes following perinatal brain injury.

## METHODS
### Study selection
The review was conducted as per the pre-registered protocol (CRD42021278572) and the Preferred Reporting Items for Systematic Reviews and Meta-Analyses (PRISMA) statement.[11] We included observational comparative studies exploring neurodevelopmental outcomes of children over 5 years of age after perinatal brain injury, published between 2000 and September 2021 (table 1). The DHSC definition of perinatal brain injuries used includes intraventricular haemorrhage (IVH), preterm white matter injury (WMI), stroke, central nervous system infection, hypoxic-ischaemic encephalopathy (HIE) and kernicterus diagnosed during the neonatal period.[6 12] We did not include seizures in isolation. For inclusion, studies were required to have a non-brain injured comparator group. The primary outcome was neurodevelopmental impairment; secondary outcomes included motor, cognitive, speech and language, behavioural and neuropsychological, visual and hearing outcomes and seizures.

A search strategy incorporating 99 key terms and mesh headings was developed in Medline Ovid, adapted and run across 10 databases (online supplemental files 1; 2). Snowballing techniques were used to augment search sensitivity. All titles were screened independently by two reviewers. The full texts of all potentially relevant titles were retrieved, reviewed and their risk of bias assessed by two trained reviewers independently (PR, CC, MV, JD and SS). Disagreements were arbitrated by a third reviewer.

### Data extraction and synthesis
Studies were stratified by brain injury type, substratified by age of outcome assessment and outcome type, and summarised in a narrative synthesis. Where sufficient suitable data were available from contextually and clinically comparable studies, data were pooled in random effects meta-analyses using RevMan V.5.4. Continuous data were pooled using the inverse variance method; dichotomous data were pooled using the Mantel-Haenszel method; and analysis data from studies which did not provide raw data were pooled with dichotomous data from other studies using the generic inverse variance method.[13] Where studies provided insufficient comparative data for a particular outcome, the combined incidence figures for that outcome within the brain injured population was calculated across studies using the Fisher's exact test for binomial data.[14] Statistical heterogeneity was assessed

using the $I^2$ statistic and substantial heterogeneity (>85%) was explored further in subgroup analyses.

### Quality assessment
The Newcastle-Ottawa Tool was used to assess risk of bias across three domains: population selection, the comparability of the 'brain injured' and 'non-brain injured' comparator groups, and outcome assessment.[15] Studies were classed as poor, fair, or good for each domain and given an overall risk of bias classification.

### Patient and public involvement
Patients or the public were not involved in the design or conduct of this review. However, the review's findings will be used to shape the larger CHERuB study in partnership with our parent advisory panel.

## RESULTS
Searches identified 14 210 records and 42 studies were included (figure 1). Studies focused on IVH (n=27), WMI among preterm infants (n=15), perinatal stroke (n=8), neonatal meningitis (n=4) and HIE (n=3); these were not mutually exclusive (online supplemental file 3). Most studies were undertaken in the USA (n=10), the UK (n=8), the Netherlands (n=5) or Australia (n=4). These were prospective (n=27) or retrospective cohort studies (n=14). Included studies were deemed to be moderate (n=17) or low risk of bias (n=27) (online supplemental file 4).

### Preterm injuries
The 29 studies exploring outcomes after IVH or WMI mostly included infants born <32 weeks' gestation (n=22) after the year 2000 (n=18) (online supplemental file 3). Most studies confirmed injury on ultrasound or MRI (n=22), these were reviewed by radiologists (n=6), neonatologists (n=3) or both (n=1); 14 studies used the Papile classification; only 2 studies stratified results by laterality.

Nine studies explored neurodevelopmental impairment at 5–14 years of age after preterm brain injury including IVH (n=9) and WMI (n=6).[16–24] Two comparable studies highlighted a considerably increased pooled crude risk of moderate-to-severe neurodevelopmental impairment after IVH grade 3–4 at 8 years of age OR 3.69 (95% CI 1.7 to 7.98; 2 studies) $I^2$=0% (figure 2, table 2).[18 21]

Six studies explored motor outcomes after IVH grades 3–4: they consistently highlighted an increased risk of motor impairment at 5–12 years of age.[21 24–28] Additionally, two comparable studies reported an eightfold higher crude risk of cerebral palsy after IVH grades 3–4 OR 8.13 (95% CI 4.64 to 14.22; 2 studies; 1557 subjects) $I^2$=0% (figure 3).

Cognitive outcomes at school age after preterm brain injuries were reported by 16 studies using 25 different cognitive assessment tools — limiting the potential for meta-analysis (online supplemental file

**Table 1** Inclusion and exclusion criteria

| Inclusion criteria | Exclusion criteria |
|---|---|
| Peer-reviewed observational studies (cohort, case–control, cross-sectional). | Non-comparative studies, opinions, commentaries, reviews, case reports, lab studies. |
| Studies in all languages. | Studies where the population includes adults and children and the data for children cannot be extracted. |
| Studies published after 2000. | Studies focused on children with IVH grades 1–2, neonatal seizures, hypoglycaemic brain injury, or neonatal abstinence syndrome. |
| Children with a diagnosis of brain injury occurring at or around the time of birth (including during the neonatal period) as defined by the DHSC (including those with any white matter injury but not including those with isolated seizures). | Studies which include infants with brain injuries diagnosed during the neonatal and infancy period where most were diagnosed outside of the neonatal period. |
| Studies including infants with moderate to severe HIE born in the post-therapeutic hypothermia era (ie, where infants received therapeutic hypothermia). | Studies including infants with moderate to severe HIE born during the pre-therapeutic hypothermia era or in low or middle income countries that do not offer therapeutic hypothermia. |
| Studies focused on school-aged neurodevelopmental outcomes (of children between 5 and 18 years of age) including:<br>Primary outcome(s):<br>Neurodevelopmental impairment, as defined by authors (including direct testing, clinical record review and parental interview/survey)<br>Secondary outcome(s):<br>1. Any cognitive impairment, as defined by authors (direct testing).<br>2. Mild cognitive impairment (intelligence or developmental quotient 1–2 SDs below the mean).<br>3. Moderate to severe cognitive impairment (intelligence or developmental quotient more than 2 SDs below the mean).<br>4 Executive dysfunction, as defined by authors (direct testing)<br>1. Low numeracy, as defined by authors (by direct testing or educational achievement tests).<br>2. Low literacy, as defined by authors (by direct testing or educational achievement tests).<br>3. Special educational needs as defined by authors (school or parental report).<br>4. Motor impairment, as defined by authors (including direct testing, clinical record review, and reporting).<br>5. Visual-motor impairment, as defined by authors (on direct testing).<br>6. Emotional-behavioural difficulty, as defined by authors (including direct testing, clinical record review, and parental reporting.<br>7. Speech and language impairment, as defined by authors (on direct testing).<br>8. Visual impairment, as defined by authors (including direct testing, clinical record review and parental reporting).<br>9. Hearing impairment, as defined by authors (including direct testing, clinical record review, and parental reporting).<br>10. Epilepsy/seizures, as defined by authors (including medical history-taking, clinical record review, and parental reporting. | Studies of infants with mild HIE. |
| | Studies reporting outcomes for children diagnosed with brain injury beyond the neonatal period. |
| | Studies where comparable outcome data from those with and without perinatal brain injury cannot be extracted. |

DHSC, Department of Health and Social Care; HIE, hypoxic-ischaemic encephalopathy; IVH, intraventricular haemorrhage.

## PRISMA 2009 Flow Diagram

**Identification**

Records identified through database searching: (n=14,210)

Additional records identified through other sources (n=8)

Records collated after deduplication (n=10,178)

**Screening**

Records screened (n=10,178)

Abstracts excluded as did not address review question (n= 8797)

**Eligibility**

Full-text articles assessed for eligibility (n=1381)

Full-text articles excluded, with reasons (n = 1339)

- Does not address review question (n=508)
- No comparative outcomes (n=298)
- Published before 2000 (n=251)
- Not peer-reviewed (n=140)
- Unable to extract outcomes of interest (n=131)
- Methodologically flawed (n=9)
- Duplicate (n=2)

**Included**

Studies included (n=42)

**Figure 1** Preferred Reporting Items for Systematic Reviews and Meta-Analyses (PRISMA) flow diagram.

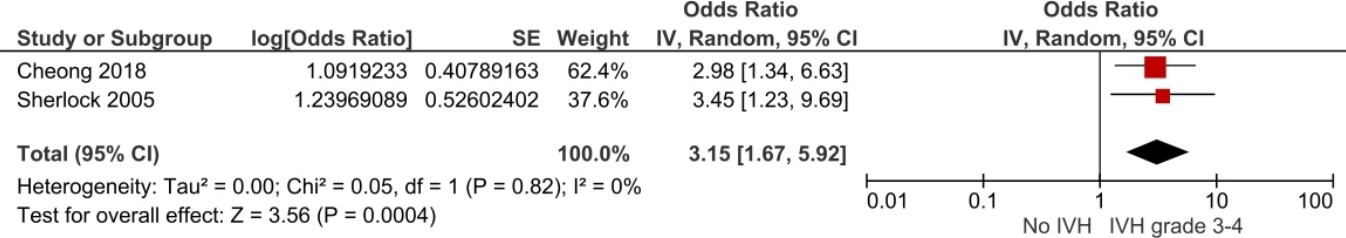

**Figure 2** Crude risk of neurodevelopmental impairment at 8 years of age after IVH grades 3–4. IV, inverse variance; IVH, intraventricular haemorrhage.

**Table 2** Overview of key findings for school-age outcomes of infants with perinatal brain injury compared with those without brain injury

| | NDI | Cognitive | Motor | Speech and language | Behavioural | Hearing† | Vision† | Other |
|---|---|---|---|---|---|---|---|---|
| **IVH grades 3–4*** | 6 studies[15 17–21] | 9 studies (15, 20, 21, 24–26, 30, 70) | 6 studies[20 23–26 33] | 3 studies[20 21 25] | 3 studies[15 24 35] | 3 studies[21 26 38] | 5 studies[15 21 26 33 38] | |
| | 2 comparable studies in meta-analysis[17 20] | Not comparable | Not comparable | Not comparable | Not comparable | Not comparable | Not comparable | |
| | **Meta-analysis (2 studies):** Increased risk of moderate–severe neurodevelopmental impairment OR 3.15 (95% CI 1.67 to 5.92) I²=0% | Consistently highlighted lower cognitive scores | All reported increased risk of motor impairment | **Van de Bor 2004:**[22] no significant difference in language scores | **Brouwer et al:**[25] no association with any behavioural domains assessed (internalising, externalising and sleep problems) | Outcome too rare for inferential analysis | Outcome too rare for inferential analysis in most studies. | |
| | **Van de Bor et al:**[22] increased prevalence of disability 31% vs 16% | **Brouwer et al:**[25] significantly lower performance IQ but preserved verbal IQ. Lower IQ for those with IVH grade 4 requiring neurosurgery (91±10 vs 98±15) but little difference for those with grade 3 IVH requiring neurosurgery (96±15 vs 98±15). | **Cerebral palsy** 3 comparable studies OR 8.67 (95% CI 5.27 to 14.28) I²=0%. | **Sherlock et al:**[21] downward trend in language scores from no brain injury to each grade of IVH but not statistically significant p=0.12 | **Adant et al:**[16] no increased risk of attention deficits, conduct issues or ASD aOR 1.24 (95% CI 0.32, 4.8). | **Kaur et al:**[39] increased risk of hospitalisation for otologic reasons HR 7.87 (95% CI 5.31 to 11.67) | **Adant et al:**[16] no increased risk of visual impairment (needing glasses) aOR 0.47 (95% CI 0.13 to 1.69) | |
| | | **Hollebrandse et al:**[26] increased risk of cognitive impairment OR 2.68 (95% CI 1.21 to 5.94). Increased risk of academic impairment across all academic domains: reading OR 3.62 (95% CI 1.59 to 8.24); spelling OR 4.48 (95% CI 1.8 to 11.2); arithmetic OR 2.79 95% CI 1.2 to 6.48) | | **Hollebrandse et al:**[26] Increased risk of impaired reading OR 3.62 (95% CI 1.59, 8.24) and spelling OR 4.48 (95% CI 1.8 to 11.2) | **Davidovich et al:**[36] no increased risk of ASD (n=10, 3.9% vs n=103, 2.2% p=0.085) | | **Klebermass-Schrehof et al:**[27] increased prevalence of visual impairment (needing glasses or blindness) after IVH grade 3 (45.4%) and IVH grade 4 (90.9%) vs comparators (7.5%). | |
| | | **Sherlock et al:**[21] significantly lower IQ scores after IVH grade 4 vs IVH 1–3 and no brain injury, also seen for several domains: freedom from distractibility, processing speed, reading, spelling and arithmetic. No difference in executive function. | | | | | **Kaur et al:**[39] increased risk of hospitalisation for ophthalmic reasons HR 7.87 (95% CI 5.31 to 11.67). | |
| | | **Van de Bor:**[22] increased special education needs at 5, 9 and 14 years aOR 3.99 (95% CI 1.36 to 11.69). | | | | | **Klebermass-Schrehof et al:**[27] significantly lower VMI scores (67.5±14 vs 76±26.8; p=0.04) | |

Continued

**Table 2** Continued

| | NDI | Cognitive | Motor | Speech and language | Behavioural | Hearing† | Vision† | Other |
|---|---|---|---|---|---|---|---|---|
| WMI* | 3 studies[16 17 22] | 4 studies (16, 29, 32, 70) | **Cerebral palsy** 1 study[16] | 1 study[29] | 4 studies (16, 35, 36, 71) | 0 studies | 1 study[32] | |
| | Not comparable | Not comparable | **Campbell 2020:** increased risk of cerebral palsy aOR 18.63 (95% CI 7.37 to 47.06) | **Jansen et al:**[30] No association between WMI and spelling (B 1.076 p=0.075) or reading performance (B 0.241 p=0.483) | Not comparable | | | |
| | **Campbell et al:**[17] living with no impairment was less common with WMI (n=12, 40%) vs controls (n=487, 76%) | **Van den Hout et al:**[33] 50% with PVL had IQ scores <85 vs 11.8% without injury and a lower performance age 4.3 years vs 6.2 years | | | Conflicting results | | | |
| | | **Campbell et al:**[17] increased risk of moderate-to-severe cognitive impairment aOR 5.07 (95% CI 2.13 to 12.02) | | | **Campbell et al:**[17] No increased risk of: ADHD (n=3, 10% vs n=97, 15%); anxiety (n=3, 10% vs n=98, 15%); depression (n=7, 23% vs n=100, 16%); or ASD aOR 0.74 (95% CI 0.09 to 5.88) | | | |
| | **Cheong 2018:**[18] increased risk of survival with major disability after cPVL aOR 9.17 (95% CI 3.57 to 23.53) | **Jansen et al:**[30] WMI predictive of poorer performance on standardised mathematics tests (B 1.856 p=0.003), but not performance on spelling (B 1.076 p=0.075) or reading tests (B 0.241 p=0.483) | | | **Davidovitch et al:**[36] No increased risk of ASD after PVL (n=5, 2.5% vs n=88, 2.3% p=0.86) | | | |
| | **Vollmer et al:**[23] Disabling impairments were more common after cPVL at <28 weeks' gestation (n=3, 75% <28 weeks) vs controls (n=3, 8%) and at over 28 weeks' gestation (n=6, 50% vs n=14, 6%) | | | | **Whitaker et al:**[37] increased risk of ADHD aOR 6.83 (95% CI 1.26 to 36.91); major depression aOR 2.59 (95% CI 1.02 to 6.58); tic disorders aOR 9.77 (95% CI 1.69 to 56.47); obsessive compulsive disorders aOR 15.32 (95% CI 1.82 to 128.74) | | | |

**Table 2** Continued

| | NDI | Cognitive | Motor | Speech and language | Behavioural | Hearing† | Vision† | Other |
|---|---|---|---|---|---|---|---|---|
| Stroke | 0 studies | 6 studies[39 41 42 44–46] 5 comparable studies in meta-analysis[39 41 44–46]<br><br>**Meta-analysis** (5 studies): significant mean difference in full scale IQ −24.2 (95% CI −30.73 to −17.67) I²=80%<br><br>**Trauner**[47] and **Gold**:[42] no significant difference in full scale IQ scores in left vs right-sided strokes<br><br>**Ballantyne et al**:[40] significantly lower performance IQ (p=0.002) and verbal IQ (p<0.0001). Lower mean scores for reading (p<0.0001), spelling (p=0.001) and arithmetic (p<0.0001) at 7–8 years persisting to 10–12 years<br><br>**Tillema et al**:[46] reduced verbal IQ scores (mean 84 SD 13.4) vs (mean 108 SD 14.2 p=0.002)<br><br>**Kolk et al**:[43] poorer attention (across 4 of the 7 assessment sub-domains), visuo-spacial function (across 4 of the 5 subdomains) and memory and learning (across 4 of the 6 subdomains), but normal executive function scores. Those with left-sided strokes had poorer neuropsychological scores.<br><br>**Northam et al**:[45] most children are in mainstream education (n=28, 93%) but many require additional support (n=12, 40%) | 5 studies[39 41–44] Combined hemiparesis incidence: 61% (95% CI 39.2% to 82.9%) I²=88%<br><br>**Kolk et al**:[43] moderate-to-severe neuromotor impairment in 62% (n=13) and significantly lower scores on 5/6 sensorimotor domains of the NEPSY | 5 studies[39 40 42 44 45] 3 comparable studies in meta-analysis<br><br>**Meta-analysis** (3 studies): lower receptive language scores−20.88 (95% CI −36.66 to −5.11) I²=88% and lower expressive language scores −20.25 (95% CI −34.36 to −6.13) I²=87%<br><br>**Ballantyne et al 2007**[41] and **Ballantyne et al 2008**[40] deficits in receptive language scores at 7–8 years persist at 10–12 years but expressive language scores improved (p=0.012) particularly for children with right-sided strokes (p=0.034)<br><br>**Kolk et al**:[43] significantly lower scores for 8/9 NEPSY domains including phonologic processing, comprehension of instructions, correct speeded naming, repetition of nonsense words, verbal fluency (semantic and phonetic), oromotor sequences and sentence comprehension | 1 study[46] | 1 study[43]<br><br>**Martin**:[44] left-sided strokes predispose children to contralateral auditory neglect and right-sided strokes predispose children to bilateral auditory neglect | 1 study[39]<br><br>**Ballantyne et al**:[40] visual field defects are common (n=7, 26%) after perinatal stroke | **Seizures** 8 studies[39 42 43 45 46]<br><br>5 comparable studies[39 42 43 45 46]<br><br>Combined incidence of seizures: 40.1% (95% CI 26.8% to 53.3%) I²=56% |

Continued

**Table 2** Continued

| | NDI | Cognitive | Motor | Speech and language | Behavioural | Hearing† | Vision† | Other |
|---|---|---|---|---|---|---|---|---|
| Meningitis | 3 studies[47–49]<br>Not comparable<br><br>All reported increased risk of neurodevelopmental impairment<br><br>**Bedford 2011:** increased prevalence of neuromotor disability (n=45, 16% vs n=2, 0.1%)<br><br>**Stevens et al:**[50] Risk of severe disability seen in Bedford 2011 at 5 years of age persisted until 9–10 years (n=12, 10.8% vs n=0, 0%)<br><br>**Horvath-Puho et al:**[49] increased risk of any neurodevelopmental impairment after GBS meningitis in the Netherlands RR 5.30 (95% CI 2.57 to 10.89) and Denmark RR 7.80 (95% CI 4.42 to 13.77) at 5 years of age persisting to 11 years in the Netherlands RR 2.99 (95% CI 1.83 to 4.88) and 15 years in Denmark RR 3.15 (95% CI 1.82 to 5.46) | 1 study[49]<br><br>**Stevens 2003:**[50] significantly lower mean cognitive scores (mean 88.8 (95% CI 85 to 92) vs mean 99.4 (95% CI 97 to 102)) | 1 study[49]<br><br>**Stevens et al:**[50] significantly higher motor impairment scores (mean 7.1 (95% CI 5.9 to 8.5) vs mean 5 (95% CI 4.3 to 5.8)) | 0 studies | 0 studies | 2 studies (49, 72)<br><br>**Martinez-Cruz 2008:** increased odds of neonatal meningitis among preterm infants with sensorineural hearing loss OR 4.37 (95% CI 1.7 to 10.9<br><br>[50]**Stevens 2003:** 3.6% (n=4) had hearing loss compared with none in the control group. | 1 study[49]<br><br>**Stevens et al:**[50] Bilateral visual impairment was common after neonatal meningitis (n=18, 17%) | |

Continued

**Table 2** Continued

| | NDI | Cognitive | Motor | Speech and language | Behavioural | Hearing† | Vision† | Other |
|---|---|---|---|---|---|---|---|---|
| HIE | 0 studies | 3 studies[30 50 51] (two of the same population)<br><br>Not comparable<br><br>Koc et al.[31] preterm infants with HIE significantly more likely to have below average IQ scores (n=8, 89% vs n=24, 30% p=0.001)<br><br>Lee-Kelland et al[51] and Tonks et al.[52] report lower full scale IQ scores after moderate to severe HIE (mean difference –13.62 (95% CI –20.53 to –6.71)) and poorer perceptual reasoning, working memory and processing speed. Children with previous HIE more likely to receive additional classroom support OR 10 (95% CI 1.16 to 86) | 2 studies[50 51] (of the same population)<br><br>Lee-Kelland et al[51] and Tonks et al.[52] significantly lower motor scores (mean difference –2.12 (95% CI –3.93 to –0.30)) after moderate-severe HIE (for children without cerebral palsy) | 2 studies[50 51] (of the same population)<br><br>Lee-Kelland et al[51] and Tonks et al.[52] significantly lower verbal comprehension scores (mean difference –8.8 (95% CI –14.25 to –3.34)) after moderate to severe HIE. | 2 studies[50 51] (of the same population)<br><br>Lee-Kelland et al[51] and Tonks et al.[52] higher behavioural difficulty scores (median score 12 IQR (6.5, 13.5 vs median score 6 IQR (2.25, 10) p=0.005) | 0 studies | 0 studies | |
| Kernicterus | 0 studies | | | | | | | |

*Does not include studies where infants with IVH grades 3–4 cannot be separated from those with WMI or those with IVH 1–2.
†Does not include studies using hearing or visual outcomes only as part of their composite outcome.
ADHD, attention-deficit/hyperactivity disorder; aOR, adjusted OR; ASD, autism spectrum disorder; cPVL, cystic PVL; GBS, group B Streptococcus; HIE, hypoxic-ischaemic encephalopathy; IVH, intraventricular haemorrhage; NDI, Neurodevelopmental impairment ; PVL, periventricular leukomalacia; RR, Risk ratio; VMI, visual motor integration; WMI, white matter injury.

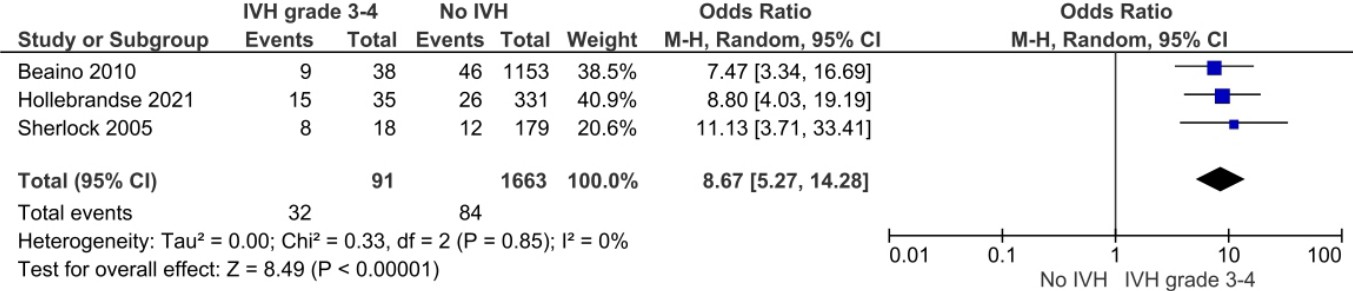

**Figure 3** Crude risk of cerebral palsy after IVH grades 3–4. IVH, intraventricular haemorrhage; M-H, Mantel-Haenszel.

3).[16 17 21 22 24–35] Educational outcomes were reported by five studies.[21 22 26 30 35]

Studies consistently reported lower cognitive scores at school age following IVH grade 3–4.[16 21 22 25 26 26 27 3 1 35] Hollebrandse *et al* reported an increased risk of cognitive impairment at 8 years of age OR 2.68 (95% CI 1.21 to 5.94).[26] van de Bor and den Ouden and Hollebrandse *et al* reported that the cognitive impact of IVH grade 3–4 affected educational needs.[22 26] van de Bor and den Ouden reported increased special educational needs at 5, 9 and 14 years: the adjusted risk at 14 years of age was marked, adjusted OR 3.99 (95% CI 1.36 to 11.69).[22] Studies reported no significant differences in language scores after IVH grades 3–4.[21 22] However, an association with reading OR 3.62 (95% CI 1.59 to 8.24), spelling OR 4.48 (95% CI 1.8 to 11.2), and arithmetic OR 2.79 (95% CI 1.2 to 6.48) impairment was demonstrated.[26] Most studies highlighted cognitive effects after WMI.[17 30 33 35]

Studies exploring behavioural outcomes after IVH 3–4 did not find any associations with attention deficits, conduct issues or autism spectrum disorder (table 2).[16 25 36] However, there was conflicting evidence around the mental health effects of WMI.[17 37]

Studies exploring hearing impairment after IVH and/or WMI were small or not comparable. Ten studies explored visual impairment after IVH or WMI, four provided meaningful outcome data.[16 21–23 27 28 33 34 38 39] An increased prevalence of visual impairment after IVH grades 3–4 (45.4% and 90.9%) compared with controls (7.5%) was reported in addition to significantly lower visual motor integration scores.[27]

## Perinatal stroke

Eight comparative studies explored school-age outcomes after perinatal stroke, these included 177 children with perinatal stroke (100 left sided and 54 right sided—not all studies specified laterality) and 232 comparator children (online supplemental file 3).[40–47] Infants' gestational age was largely unspecified. Five studies presented a combined incidence of childhood seizures after perinatal stroke of 40.1% (95% CI 26.8% to 53.3%; 5 studies; 115 subjects) $I^2$=56% (online supplemental file 5).[40 43 44 46 47] The combined incidence of hemiparesis after perinatal stroke was 61% (95% CI 39.2% o 82.9%, $I^2$=88%). There was considerable heterogeneity across studies, and likely detection bias (online supplemental file 6).[40 42–45]

Five studies identified a significant combined mean difference in full scale IQ scores at 7–13 years of age after perinatal stroke: −24.2 (95% CI −30.73 to −17.67; 5 studies; 296 subjects) $I^2$=80% (figure 4).[40 42 45–47] There was heterogeneity across studies in terms of assessment timing, assessment tools and combining those with left-sided and right-sided strokes.

Differences in stroke laterality partially explained the heterogeneity. The combined mean difference in full scale IQ following left-sided strokes was −26.01 (95% CI −29.1 to −22.93; 2 studies; 113 subjects) $I^2$=0%; compared with −26.7 (95% CI −39.38. to -14.02; 2 studies; 99 subjects) $I^2$=76% for right-sided strokes. No significant differences in cognitive outcomes were found by laterality.[40 42 45–47]

Kolk *et al* reported significantly lower scores across all NEPSY domains other than executive function after

**Figure 4** Pooled mean difference in IQ scores at 7–13 years between those with and without perinatal stroke. IV, inverse variance.

perinatal stroke, including attention, visuospacial function, memory and learning.[43]

Two studies presented educational outcomes after perinatal stroke. Although Northam *et al* found that most children with perinatal stroke were in mainstream education (n=28, 93%), they also highlighted that additional educational support was often required (n=12, 40%). This was in keeping with Ballantyne *et al*[40] reporting lower mean scores for reading (85 (16.1) vs 113 (13.3); p<0.0001), spelling (82.5 (18.2) vs 106.2 (15.9) p=0.001) and arithmetic (91.5 (10.2) vs 111.9 (11.2) p<0.0001) after perinatal stroke compared with controls at 7–8 years of age, persisting on re-assessment at 10–12 years.

Kolk *et al* reported significantly lower scores compared with controls across most NEPSY language domains following perinatal stroke.[43] Significantly lower receptive and expressive mean language scores on the CELF assessment were also reported across studies: –20.88 (95% CI –36.66 to –5.11; 2 studies; 137 subjects) $I^2$=88% and –20.25 (95% CI –34.36 to –6.13; 2 studies; 137 subjects) $I^2$=87%, respectively (online supplemental files 7, 8).[40 45] Statistical heterogeneity may have been as a result of studies combining left-sided and right-sided strokes and the varying age of outcome assessment. Studies highlighted that deficits in receptive language scores present at 7–8 years persisted at 10–12 years but that expressive language scores improved (p=0.012).[40 41]

### Meningitis

Studies consistently reported an increased risk of neurodevelopmental impairment after neonatal meningitis (table 2).[48–50] An increased likelihood of neuromotor disability at 5 years of age (n=45/274, 16%) compared with controls (n=2/1391, 0.1%) was reported (online supplemental file 3).[48] On reassessment of the same population at 9–10 years, this increased risk of severe disability persisted (n=12, 10.8% compared with n=0, 0%).[50] An increased risk of any neurodevelopmental impairment at 5 years after neonatal group B *Streptococcal* meningitis was also reported in the Netherlands, RR 5.30 (95% CI 2·57 to 10·89), and in Denmark, RR 7.80 (95% CI 4·42 to 13·77).[49] This increased risk persisted on subsequent assessment: at 11 years of age in the Netherlands, RR 2.99 (95% CI 1.83 to 4.88) and at 15 years of age in Denmark RR, 3.15 (95% CI 1.82 to 5,46).[49]

### Hypoxic-ischaemic encephalopathy

Two comparative studies (of the same cohort) explored outcomes of term-born infants with moderate-to-severe HIE, but without cerebral palsy, at school age (online supplemental file 3).[51 52] They highlighted significantly lower full scale IQ scores after HIE (mean difference –13.62 (95% CI –20.53 to –6.71)).[51] This difference in cognition was also seen for perceptual reasoning, working memory and processing speed. Children with HIE were also more likely than controls to receive additional classroom support: OR 10 (95% CI 1.16 to 86) although the CI for this risk estimate was wide.[51] Children with HIE (without cerebral palsy) also had significantly lower motor scores (mean difference –2.12 (95% CI –3.93 to –0.30)) and verbal comprehension scores (mean difference –8.8 (95% CI –14.25 to –3.34)).[51] They were also noted to have higher behavioural difficulty scores especially for emotional problems.[51]

## DISCUSSION

This review brings together the existing evidence on the later childhood outcomes of infants with perinatal brain injury. Although 42 studies are included, small study populations, limited data on injury severity and laterality, and the heterogeneity of studies limited the potential power of results. However, studies demonstrate a threefold higher risk of moderate-to-severe neurodevelopmental impairment at school age following IVH grades 3–4. Studies consistently report cognitive impairment after IVH grades 3–4 but suggest that speech and language is relatively preserved. A higher risk of hemiplegia, cognitive impairment and poorer academic performance after perinatal stroke is reported in addition to poorer receptive and expressive language scores. Studies report a higher risk of persisting neurodevelopmental impairment after neonatal meningitis — however, few studies address this question. Few comparative studies explore school-age outcomes after HIE.

In following our a priori protocol, only comparative studies were included. This was with a view to enabling inferential analyses and adjustment for key confounders such as gestation. Unfortunately due to this strict inclusion criterion, many pertinent non-comparative studies were excluded. Additionally, our searches were conducted in September 2021, more recent studies would therefore have been missed.

Heterogeneity in terms of outcomes assessed, outcome assessment tools, and timing of outcome assessment limited the comparability of studies and the potential for meta-analyses. Several meta-analyses included low numbers of studies, reducing the reliability of the $I^2$ statistic.[53] This review was also limited by the size of available studies and how studies presented data for extraction. Few studies presented adjusted data or explored childhood trajectories after perinatal brain injury.

Previous reviews were limited by a lack of comparable studies, heterogeneity, the inclusion of much older cohorts or by the inclusion of non-comparative studies.[4 54–56] While this review was also limited by studies' heterogeneity and the quality of available data, new and important findings — for example, the risk of neurodevelopmental impairment at school age after IVH 3–4 — were identified. Our finding of a higher risk of cerebral palsy after IVH grade 3-4 and motor impairments after preterm brain injuries is echoed by previous studies.[54 55 57]

Lynch and Nelson highlight that 60% of infants have neurological sequelae that emerge over time following perinatal stroke. This was in-keeping with our findings of a higher risk of hemiparesis, cognitive impairment

and speech and language impairment.[58] Several non-comparative population-based studies also mirror these findings.[59–62]

Although previous reviews highlight an increased risk of various neurodevelopmental impairments after neonatal meningitis in early childhood — we are unaware of any focusing on school-age outcomes after neonatal meningitis.[4 63]

The review's findings of potential ongoing impairments across cognitive, speech and language, and behavioural domains — in addition to a need for increased school support — after HIE are mirrored by other studies.[64–68] Shankaran *et al* and Azzopardi *et al* highlight ongoing neurodevelopmental sequelae at school age among children who received therapeutic hypothermia for moderate to severe HIE.[64 65 67]

## Implications

Considerable gaps in the evidence are highlighted, particularly around the risk of specific outcomes following different types of injury, the precision around risk estimates, the impact of different factors (such as injury laterality) and the developmental trajectories of these children. This information is key to prepare families for the future, inform enhanced developmental surveillance, and enable targeted multidisciplinary support to help affected children to reach their full potential. As such, this review highlights a pressing need for high-quality, comparative studies which use the 'Core Outcomes In Neonatology' to explore long-term outcomes after perinatal brain injury and permit future meta-analyses.[10] Additionally, to meet the DHSC ambition to reduce perinatal brain injury, real-time longitudinal population data, extending beyond the neonatal period to childhood, are needed. This could be achieved through linkage of existing population datasets within the UK which is a key objective of the CHERuB study.

## CONCLUSION

This review provides an overview of existing evidence of the impact of perinatal brain throughout childhood. Studies' heterogeneity significantly limited the potential for evidence synthesis.

**Author affiliations**
[1]Population Policy and Practice, University College London Great Ormond Street Institute of Child Health, London, UK
[2]Department of Primary Care Health Sciences, University of Oxford, Oxford, UK
[3]Department of Paediatrics, Cambridge University Hospitals NHS Foundation Trust, Cambridge, UK
[4]Paediatric Intensive Care Unit, Great Ormond Street Hospital for Children NHS Foundation Trust, London, UK
[5]King's College London, London, UK
[6]Neonatal Intensive Care Unit, Chelsea and Westminster Hospital NHS Foundation Trust, London, UK
[7]Neonatal Medicine, School of Public Health, Faculty of Medicine, Imperial College London, London, UK

**Contributors** PR conceptualised and designed the review, reviewed and appraised studies, undertook data extraction and synthesis, drafted the initial manuscript, and reviewed and revised the manuscript and is the content guarantor. CC conceptualised and designed the review, designed and oversaw the search strategy, reviewed and appraised studies, undertook data extraction, and reviewed and revised the manuscript. KC, MV, JD, SS and FH reviewed and appraised studies, undertook data extraction, and reviewed and revised the manuscript. JG was the lead statistician for the review, he advised on and oversaw the data analysis, and reviewed and revised the manuscript. CB, CG and AS oversaw and supervised the review and critically revised the manuscript for important intellectual content. All authors approve the final manuscript as submitted and agree to be accountable for all aspects of the work.

**Funding** This review was supported by an NIHR Doctoral Fellowship award (NIHR301457).

**Competing interests** CG is funded by the UK Medical Research Council (MRC) through a Transition Support Award. In the past 5 years, he has received support from Chiesi Pharmaceuticals to attend an educational conference and has been investigator on received research grants from the Medical Research Council, National Institute of Health Research, Canadian Institute of Health Research, Department of Health in England, Mason Medical Research Foundation, Westminster Medical School Research Trust and Chiesi Pharmaceuticals. CB is funded by the UK National Institute of Health Research (NIHR) Advanced Fellowship Award.

**Patient and public involvement** Patients and/or the public were not involved in the design, or conduct, or reporting, or dissemination plans of this research.

**Patient consent for publication** Not applicable.

**Provenance and peer review** Not commissioned; externally peer reviewed.

**Data availability statement** All data relevant to the study are included in the article or uploaded as online supplemental information.

**ORCID iDs**
Philippa Rees http://orcid.org/0000-0002-1074-5837
Fergus Harnden http://orcid.org/0000-0001-6151-3406
Cheryl Battersby http://orcid.org/0000-0002-2898-553X
Chris Gale http://orcid.org/0000-0003-0707-876X

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
