## [Reviewer comments · BMJ Paediatrics Open]

This paper was submitted to a another journal from Archives of Disease in Childhood but declined for publication following peer review. The authors addressed the reviewers' comments and submitted the revised paper to BMJ Paediatrics Open. The paper was subsequently accepted for publication at BMJ Paediatrics Open.

ARTICLE DETAILS

TITLE (PROVISIONAL)	School-age outcomes of children after perinatal brain injury: a systematic review and meta-analysis
AUTHORS	Rees, Philippa Callan, Caitriona Chadda, Karan Vaal, Meriel Diviney, James Sabti, Shahad Harnden, Fergus Gardiner, Julian Battersby, Chery Gale, Chris Sutcliffe, Alastair

VERSION 1 – REVIEW

REVIEWER	Reviewer name: Dr. Seetha Shankaran Institution and Country: Wayne State UniversityChildrens Hospital of MI, United States Competing interests: None
REVIEW RETURNED	14-Nov-2022

GENERAL COMMENTS	1) Because of the requirements of this meta-analysis that infants have to have the brain injury condition and be compared to infants without the brain injury, selection of articles that are relevant is very limited. 2) Why select a cohort that is so rare and so unusual as neonates with or without spontaneous intestinal perforation with or without brain hemorrhage? This really limits usefulness of this evaluation. 3) Some articles selected have very small numbers of infants with the diagnosis in question; reference 17, 19, 21, 23 and 25 with perinatal brain injury. 4) Under hypoxic-ischemic encephalopathy (HIE) the articles selected have sample sizes or 9 and 29 subjects? 5) In the studies examining intracranial hemorrhage some have cranial sonograms read by neonatologists; Sonograms should be read by radiologists. 6) The findings of studies of preterm infants are not representative of the extremely low birth weight infants. Infants <28 weeks' gestation are the ones at highest risk for impaired outcomes. 7) Studies showing impact of brain injury on higher cognitive functions are very small ones.
---

	8) Difficult to evaluate impact of stroke studies without knowing gestational age of infants in all studies. 9) Two of the stroke studies with small sample sizes report on an array of outcome, data is underpowered for these analyses. 10) Authors have dismissed as “underpowered to evaluate outcomes” and “attrition” for the RCTs of hypothermia for HIE that had childhood outcomes described in detail. The Coolcap trial childhood outcome was a questionnaire and very low follow-up rate, but the NICHD and the TOBY RCTs had excellent follow-up rates and a wealth of data on outcomes, also see Pappas et al on cognitive trajectories after hypothermia for HIE. 11) It is suggested that merging data from 2 longitudinal programs would provide information on childhood outcomes after perinatal brain insult. Do these cited projects have the data collection capabilities that would provide needed information?
--	--

VERSION 1 – AUTHOR RESPONSE

Editor comments

The authors are all outstanding and have long-standing history of contributing greatly to the literature. In addition, this was clearly quite an undertaking. However, it is of concern that a single meta-analysis of complex school age outcomes for a wide range perinatal brain injury may be far too all encompassing, particularly given the differences in the biologic pathways of each injury at differing gestational ages. Similarly, the range of school age assessments is quite different. Some significant studies appear to be missing, although the authors may have specific reasons for their lack of inclusion.

Response: Thank you for these comments. We agree that the definition of perinatal brain injury is wide-ranging, however this is the standardised and agreed definition of perinatal brain injury developed by the Department of Health. The definition of perinatal brain injury and the consensus methods used in its development and agreement were previously published in Archives of Disease in Childhood – Fetal and Neonatal edition. We therefore undertook this review to explore whether the conditions included within this definition of perinatal brain injury resulted in childhood morbidity. Additionally we undertook separate meta-analyses for each condition rather than a single meta-analysis.

We highlight this in the introduction:

“... the Department of Health and Social Care (DHSC) has committed to halving the rate of perinatal brain injuries by 2030 as part of the national maternity ambition.(5) To monitor progress towards this goal, a standardised definition of perinatal brain injury was developed. This definition – which encompasses moderate to severe Hypoxic Ischaemic Encephalopathy (HIE), perinatal stroke, central nervous system infections (CNS), kernicterus, intraventricular haemorrhage (IVH) grade 3-4, and cystic periventricular leukomalacia – includes ‘indicators’ of such injuries during the neonatal period.(6) The degree to which this definition captures and represents true perinatal brain injuries is unclear and requires us to look beyond the neonatal period.(6) Focusing on the childhood outcomes of infants with perinatal brain injury provides a fuller understanding of the population captured by the DHSC definition.” (Page 4; Para 1; Line 3)

The heterogeneity of included studies (in terms of the range of school assessments used to determine the same type of neurodevelopmental outcome) is indeed problematic and one of the main findings of this review that we believe is important to disseminate. We highlight this in the study conclusion.

“Considerable gaps in the evidence are highlighted and studies’ heterogeneity significantly limited the potential for evidence synthesis.”[Page 14; Para 2; Line 2]

We had strict inclusion and exclusion criteria, particularly around requiring a non-brain injured comparator group, therefore several studies not meeting these criteria were excluded. We have highlighted this in the limitations section.

“Due to our strict inclusion criteria (especially requiring a non-brain injured comparator group) many pertinent studies were excluded.”[Page 12; Para 2; Line 3]

Reviewer comments

Comment 1: Because of the requirements of this meta-analysis that infants have to have the brain injury condition and be compared to infants without the brain injury, selection of articles that are relevant is very limited.

Response 1:

We agree. It is unfortunate that so few comparative studies address this question which is of great importance to parents. We highlight this in the discussion section:

“Considerable gaps in the evidence are highlighted, particularly around the risk of specific outcomes following different types of injury, the precision around risk estimates, the impact of different factors (such as injury laterality), and the developmental trajectories of these children i.e. whether outcomes are fixed, deteriorate, or improve over time. This information is key to prepare families for the future, inform enhanced developmental surveillance, and enable targeted multidisciplinary support to help affected children to reach their full potential.”[Page 14; Para 1; Line 1]

Comment 2: Why select a cohort that is so rare and so unusual as neonates with or without spontaneous intestinal perforation with or without brain hemorrhage? This really limits usefulness of this evaluation.

Response 2:

We followed an a priori protocol and therefore included all studies meeting our inclusion criteria.

Comment 3: Some articles selected have very small numbers of infants with the diagnosis in question; reference 17, 19, 21, 23 and 25 with perinatal brain injury.

Response 3:

We agree. Several included studies had small population sizes. One of the main purposes of this review was to highlight the gaps in the evidence. We now acknowledge this more clearly in the discussion.

“Heterogeneity in terms of outcomes assessed, outcome assessment tools, and timing of outcome assessment limited the comparability of studies and the potential for meta-analyses. This review was also limited by the size of available studies and how studies presented data for extraction. Few studies presented adjusted data or explored childhood trajectories after perinatal brain injury.” [Page 12; Para 2; Line 5]

“Considerable gaps in the evidence are highlighted, particularly around the risk of specific outcomes following different types of injury, the precision around risk estimates, the impact of different factors (such as injury laterality), and the developmental trajectories of these children i.e. whether outcomes are fixed, deteriorate, or improve over time. [Page 14; Para 1; Line 1]

Comment 4: Under hypoxic-ischemic encephalopathy (HIE) the articles selected have sample sizes of 9 and 29 subjects?

Response 4: Yes, unfortunately few comparative studies explored the school-aged outcomes of infants with moderate-severe HIE compared to those without HIE.

We now more clearly acknowledge this limitation in the discussion.

“This review was also limited by the size of available studies and how studies presented data for extraction. Few studies presented adjusted data or explored childhood trajectories after perinatal brain injury.” [Page 12; Para 2; Line 7]

Comment 5: In the studies examining intracranial hemorrhage some have cranial sonograms read by neonatologists; Sonograms should be read by radiologists.

Response 5: We agree. Unfortunately as with all systematic reviews we can only present the evidence available within the literature and the data disclosed by the included studies. We have more clearly highlighted this in the discussion.

“This review was also limited by the size of available studies and how studies presented data for extraction.” [Page 12; Para 2; Line 7]

Comment 6: The findings of studies of preterm infants are not representative of the extremely low birth weight infants. Infants <28 weeks' gestation are the ones at highest risk for impaired outcomes.

Response 6: We agree but as with all systematic reviews we can only present and analyse the data provided by included studies. These limitations are presented in the risk of bias table and have now been highlighted in the discussion section.

“This review was also limited by the size of available studies and how studies presented data for extraction. Few studies presented adjusted data or explored childhood trajectories after perinatal brain injury.” [Page 12; Para 2; Line 7]

Comment 7: Studies showing impact of brain injury on higher cognitive functions are very small ones.

Response 7: We agree but we are limited by the available studies meeting the review inclusion criteria. This has now been highlighted in the discussion.

“This review was also limited by the size of available studies and how studies presented data for extraction. Few studies presented adjusted data or explored childhood trajectories after perinatal brain injury.” [Page 12; Para 2; Line 7]

Comment 8: Difficult to evaluate impact of stroke studies without knowing gestational age of infants in all studies

Response 8: We agree. We are unfortunately limited by the data presented by included studies. We acknowledge this in the limitations section.

“This review was also limited by the size of available studies and how studies presented data for extraction. Few studies presented adjusted data or explored childhood trajectories after perinatal brain injury.” [Page 12; Para 2; Line 7]

Comment 9: Two of the stroke studies with small sample sizes report on an array of outcome, data is underpowered for these analyses

Response 9: We agree but are limited by the data presented by included studies. We acknowledge this in the limitations section.

“This review was also limited by the size of available studies and how studies presented data for extraction. Few studies presented adjusted data or explored childhood trajectories after perinatal brain injury.” [Page 12; Para 2; Line 7]

Comment 10: Authors have dismissed as “underpowered to evaluate outcomes” and “attrition” for the RCTs of hypothermia for HIE that had childhood outcomes described in detail. The Coolcap trial childhood outcome was a questionnaire and very low follow-up rate, but the NICHD and the TOBY RCTs had excellent follow-up rates and a wealth of data on outcomes, also see Pappas et al on cognitive trajectories after hypothermia for HIE.

Response 10: Thank you for highlighting this. We certainly did not intend to dismiss the importance of these seminal trials. Unfortunately they were not powered to explore longer term childhood outcomes (even with 100% follow-up) but we agree that they presented a wealth of useful data on childhood outcomes after therapeutic hypothermia. We have included reference to this in the discussion section and removed the reference to attrition on follow-up. The text now reads:

“Shankaran 2012 and Azzopardi 2014 highlight on-going neurodevelopmental sequelae at school age amongst children who received therapeutic hypothermia for moderate-severe HIE.³⁻⁵ Unfortunately these studies were not powered to explore individual (non-composite) developmental outcomes or school-age outcomes.” [Page 13; Para 4; Line 3]

Comment 11: It is suggested that merging data from 2 longitudinal programs would provide information on childhood outcomes after perinatal brain insult. Do these cited projects have the data collection capabilities that would provide needed information?

Response 11: Thank you for this comment. We believe that by linking routine data within the UK many of the gaps in evidence highlighted by this review could be addressed. This systematic review is the first-step in an NIHR-funded study aiming to do this. We have highlighted the potential of utilising these routine datasets within the discussion section.

“Additionally, to meet the DHSC ambition to reduce perinatal brain injury, real-time longitudinal population data, extending beyond the neonatal period to childhood, are necessary as the current definition is limited to ‘indicators’ of injury from the neonatal period. This could be achieved through linkage of existing population datasets within the UK and would enable monitoring of progress towards the DHSC goal and evaluation of the impact of national Quality Improvement efforts targeting perinatal brain injuries.” [Page 14; Para 1; Line 9]

VERSION 2 – REVIEW

REVIEWER	Reviewer name: Dr. Peter Flom Institution and Country: Peter Flom Consulting, United States Competing interests: none
REVIEW RETURNED	13-Dec-2022

GENERAL COMMENTS	I confine my remarks to statistical and methodological aspects of this paper. The general approach is fine, but I do have some issues to resolve before I can recommend publication. General: Note that I² can be biased when there are a small number of studies, as is the case here. See e.g. https://bmcmmedresmethodol.biomedcentral.com/articles/10.1186/s12874-015-0024-z p 3 line 29-32 The results for expressive and receptive language are identical. This must be a typo. Also, -20,25 what? What are the units? If it is a score on some instrument, then it is almost meaningless unless (like IQ) the score is well known. p. 6 lines 47-49 What were the various types?
---

	Results: It would be nice to give the number of subjects in addition to the number of studies in each set of parentheses. This would give a better sense of the amount of data. p. 11 Whenever a p value is given, some sort of effect size should also be given. E.g. for language scores, give means and sds. (also, the language scores here are not duplicated the way they are in the abstract, so that seems like a cut and paste problem). For all the forest plots: While what the authors did isn't wrong, the large amount of text makes the actual figure tiny. One way to improve this is to make them landscape format. Another is to limit number of decimal places (in the first figure) Peter Flom
--	--

VERSION 2 – AUTHOR RESPONSE

Editor in Chief comments

Comment 1: Abstract needs to include date of the search.

Response 1: Thank you for highlighting this. We have now included the dates within the abstract, which reads:

“A systematic review and meta-analyses were undertaken of studies published between 2000-September 2021 exploring school-aged neurodevelopmental outcomes of children after perinatal brain injury compared to those without perinatal brain injury.” [Line 1; Para 2; Page 2]

Comment 2: Clarify your definition of perinatal brain injury used in your search. Is it an official DHSC definition or the working Group definition developed by some of the authors? It should be detailed in the Methods.

Response 2: Apologies for not including this in the initial submission. We have now included the definition within the methods. We used the official Department of Health and Social Care definition for perinatal brain injury that was developed by a working group led by Professor Gale (on the request of the DHSC). For improved clarity we have now referenced both the DHSC publication and the academic publication specifying how the definition was developed and agreed. The methods now reads:

“The DHSC definition of perinatal brain injuries was used, which includes intraventricular haemorrhage (grade 3-4), preterm white matter injuries, stroke, central nervous system infection, moderate-severe hypoxic ischaemic encephalopathy, and kernicterus.^{1,2}” [Line 4; Para 1; Page 5]

Comment 3: Only including studies that include controls is a MAJOR limitation and needs to be stated as such in the Discussion.

Response 3: Thank you for highlighting this. We were unable to retrospectively change this criterion as it was specified in our protocol but it was indeed unfortunate that so few comparative studies addressed this question, limiting the findings.

We have amended the discussion to state this more clearly, provided a better explanation of our rationale, and referenced throughout the discussion many of the pertinent non-comparative studies that were excluded.

"In following our a priori protocol only comparative were included. This was with a view to enabling inferential analyses and permitting adjustment for key confounders such as gestation. Unfortunately due to this strict inclusion criterion many pertinent non-comparative studies were excluded." [Line 1; Para 2; Page 12]

Comment 4: You need to be far more cautious in your conclusions. The gaps in the evidence may relate to your search strategy excluding many studies.

Response 4: Thank you. We have amended the conclusion to be more cautious and removed mention of the gaps in evidence. The conclusion now reads:
"This review provides an overview of existing evidence of the impact of perinatal brain throughout childhood. Studies' heterogeneity significantly limited the potential for evidence synthesis." [Line 1; Para 2; Page 14]

Comment 5: Results Supplementary Table 6 and the Prisma Flow Diagram need to be in the main paper

Response 5: These have now been included in the main paper as Table 2 and Figure 1 respectively.

Comment 6: Discussion 2nd para Delete the first two sentences "This is the first systematic review to focus on school-age outcomes after perinatal brain injury using the DHSC definition.(6) An extensive search strategy was employed alongside a rigorous review process". Journal policy is for authors to avoid describing their study as the first and the second sentence is just you praising yourselves.

Response 6: These sentences have now been deleted.

Associate Editor comments

Comment 1: The definition on perinatal brain injury, and this definition is currently applied to assess neurological outcome at school age. That's likely useful, but does this mean that the definition depends on the outcome? Somehow, this does not appear logical. Are 'true' perinatal brain injuries limited to only 'positive' indicators (associated with poorer longterm outcome?)

Response 1: Thank you for highlighting this potential confusion. The DHSC definition is limited to 'indicators' of injuries from the neonatal period such as a positive CSF culture in neonatal meningitis. We were interested in the childhood outcomes of infants such as these as not all of them will go on to have on-going impairments as a result of this neonatal diagnosis. For improved clarity we have now included more information about the definition and referenced both the DHSC and the academic publication which provides more information on how the definition was developed and agreed (including its limitations). We have also highlighted the limitations of the definition in the discussion.

"The DHSC definition of perinatal brain injuries was used which includes intraventricular haemorrhage (grade 3-4), preterm white matter injuries, stroke, central nervous system infection, moderate-severe hypoxic ischaemic encephalopathy, and kernicterus diagnosed during the neonatal period.^{1,2} We did not include seizures in isolation." [Line 4; Para 1; Page 5]

“...to meet the DHSC ambition to reduce perinatal brain injury, real-time longitudinal population data, extending beyond the neonatal period to childhood, are necessary as the current definition is limited to ‘indicators’ of injury from the neonatal period. [Line 6; Para 1; Page 14]

Comment 2: Only including studies that include controls is perhaps too much driven by methods. Valuable data on outcome are reported on cohort studies, so that this ‘overfocus’ on methods does result in a reductionism of the available information.

Response 2: Thank you for highlighting this. We agree, it’s unfortunate that so few comparative studies were identified for inclusion. Due to our a priori protocol specifying the inclusion of comparative studies only (in order to account for the impact of key confounders such as gestation on outcomes) our hands were tied. We have amended the discussion to more clearly highlight this limitation and referenced many of the key non-comparative studies that were excluded throughout the discussion.

“In following our a priori protocol only comparative studies were included. This was with a view to enabling inferential analyses and adjustment for key confounders such as gestation. Unfortunately due to this strict inclusion criterion many pertinent non-comparative studies were excluded.” [Line 1; Para 2; Page 12]

Reviewer comments

Comment 1: General: Note that I^2 can be biased when there are a small number of studies, as is the case here.

Response 1: Thank you for highlighting this. We have now highlighted this within the limitations section of the discussion.

“Heterogeneity in terms of outcomes assessed, outcome assessment tools, and timing of outcome assessment limited the comparability of studies and the potential for meta-analyses. Several meta-analyses included low numbers of studies, reducing the reliability of the I^2 statistic.(53).” [Line 1; Para 3; Page 12]

Comment 2: p 3 line 29-32 The results for expressive and receptive language are identical. This must be a typo. Also, -20,25 what? What are the units? If it is a score on some instrument, then it is almost meaningless unless (like IQ) the score is well known.

Response 2: Thank you for highlighting this. We have amended the typographical error in the abstract and included more information on the language assessment tool used to determine mean language scores in both the abstract and the results section. The CELF is a well-recognised measure of language (much like the IQ score).

“Perinatal stroke was also associated with poorer academic performance; and lower mean receptive -20.88 (95%CI: -36.66, -5.11) and expressive language scores -20.25 (95%CI: -34.36, -6.13) on the CELF assessment.” [Line 6; Para 3; Page 2- Abstract]

“Significantly lower receptive and expressive mean language scores on the CELF assessment were also reported across studies: -20.88 (95%CI: -36.66, -5.11; 2 studies; 137 subjects) $I^2=88\%$ and -20.25 (95%CI: -34.36, -6.13; 2 studies; 137 subjects) $I^2=87\%$ respectively (Supplement 7, 8).^{3,4}” [Line 2; Para 3; Page 10]

Comment 3: p. 6 lines 47-49 What were the various types?

Response 3: Apologies but we're not clear what part of the manuscript this comment refers to. If this refers to the types of brain injuries we have now included more information in the introduction on the different brain injury types.

"The DHSC definition of perinatal brain injuries was used, which includes intraventricular haemorrhage (grade 3-4), preterm white matter injuries, stroke, central nervous system infection, moderate-severe hypoxic ischaemic encephalopathy, and kernicterus.^{1,2}" [Line 4; Para 1; Page 5]

Comment 4: Results: It would be nice to give the number of subjects in addition to the number of studies in each set of parentheses. This would give a better sense of the amount of data.

Response 4: We agree and have now included this information within the results section for all meta-analyses (where available).

Comment 5: p. 11 Whenever a p value is given, some sort of effect size should also be given. E.g. for language scores, give means and sds.

Response 5: Thank you for highlighting this oversight. We have now included additional information on effect sizes within the relevant part of the results section, which reads:

"This was in keeping with Ballantyne 2008 reporting lower mean scores for reading (85 (16.1) vs. 113 (13.3); $p < 0.0001$), spelling (82.5 (18.2) vs. 106.2 (15.9) $p = 0.001$) and arithmetic (91.5 (10.2) vs. 111.9 (11.2) $p < 0.0001$) after perinatal stroke compared to controls at 7-8 years of age, persisting on re-assessment at 10-12 years." [Line 3 Para 2 Page 10]

Comment 6: For all the forest plots: While what the authors did isn't wrong, the large amount of text makes the actual figure tiny. One way to improve this is to make them landscape format.

Response 6: Thank you for highlighting this. The plots have been generated as per the journal guidance and we therefore hope that they are easy to re-format as needed for the journal prior to publication. Of course we're more than happy to amend in any way that's helpful to the editorial team.